SOFTWARE

# AnophelesModel: An R package to interface mosquito bionomics, human exposure and intervention effects with models of malaria intervention impact

**Monica Golumbeanu**[1,2]\*, **Olivier Briët**[1,2], **Clara Champagne**[1,2], **Jeanne Lemant**[1,2], **Munir Winkel**[1,2], **Barnabas Zogo**[3], **Maximilian Gerhards**[1,2], **Marianne Sinka**[4], **Nakul Chitnis**[1,2], **Melissa Penny**[1,2,5,6], **Emilie Pothin**[1,2], **Tom Smith**[1,2]

**1** Swiss Tropical and Public Health Institute (Swiss TPH), Allschwil, Switzerland, **2** University of Basel, Basel, Switzerland, **3** University of Montpellier, Montpellier, France, **4** Department of Biology, University of Oxford, Oxford, United Kingdom, **5** The Kids Research Institute Australia, Nedlands, WA, Australia, **6** Centre for Child Health Research, University of Western Australia, Crawley, WA, Australia

\* monica.golumbeanu@swisstph.ch

**Data Availability Statement:** AnophelesModel is open source, available under a GPL 3.0 license. The

## Abstract

In recent decades, field and semi-field studies of malaria transmission have gathered geographic-specific information about mosquito ecology, behaviour and their sensitivity to interventions. Mathematical models of malaria transmission can incorporate such data to infer the likely impact of vector control interventions and hence guide malaria control strategies in various geographies. To facilitate this process and make model predictions of intervention impact available for different geographical regions, we developed AnophelesModel. AnophelesModel is an online, open-access R package that quantifies the impact of vector control interventions depending on mosquito species and location-specific characteristics. In addition, it includes a previously published, comprehensive, curated database of field entomological data from over 50 *Anopheles* species, field data on mosquito and human behaviour, and estimates of vector control effectiveness. Using the input data, the package parameterizes a discrete-time, state transition model of the mosquito oviposition cycle and infers species-specific impacts of various interventions on vectorial capacity. In addition, it offers formatted outputs ready to use in downstream analyses and by other models of malaria transmission for accurate representation of the vector-specific components. Using AnophelesModel, we show how the key implications for intervention impact change for various vectors and locations. The package facilitates quantitative comparisons of likely intervention impacts in different geographical settings varying in vector compositions, and can thus guide towards more robust and efficient malaria control recommendations. The AnophelesModel R package is available under a GPL-3.0 license at https://github.com/SwissTPH/AnophelesModel.

package R code is available online at https://github.com/SwissTPH/AnophelesModel. The R package has a dedicated website at https://swisstph.github.io/AnophelesModel/index.html which includes a comprehensive, online documentation covering the installation of the package in R, the description of all its functions, as well as detailed examples about its components and use cases available at: https://swisstph.github.io/AnophelesModel/articles/AnophelesModel.html. This documentation guides the user step-by-step along the process of using the package. All the code used to generate the results and figures presented in the manuscript is also provided online on GitHub at https://github.com/SwissTPH/AnophelesModel/tree/main/extdata.

**Funding:** This work was supported in part by the Bill & Melinda Gates Foundation, United States (https://www.gatesfoundation.org/) with grant OPP1032350 awarded to TS, and grants INV-030449 and INV-068864 awarded to EP. NC also acknowledges funding from the Bill and Melinda Gates Foundation for this work, with grants OPP#1032350 and INV-025569. The funders did not play any role in the study design, data collection and analysis, decision to publish, or preparation of the manuscript.

**Competing interests:** The authors have declared that no competing interests exist.

## Introduction

Vector control targeting *Anopheles* (*An.*) mosquitoes and protecting people from their dangerous, malaria-infectious bites has been the predominant way of reducing the malaria burden worldwide [1]. Over 220 million insecticide-treated nets (ITNs), the most common vector control tool, were distributed in 2021 [2], but the impact of these and other vector control interventions varies geographically depending on multiple factors. These factors include intra and inter-species heterogeneity in the characteristics of the vectors and geographical variation in vector species composition. *Anopheles* mosquitoes have a complex life-cycle, continuously adapting to and evolving with the surrounding environment. The species native to Africa can be very different to those found elsewhere [3]. The interactions of circadian mosquito biting patterns and the behavioural patterns of humans are particularly relevant for the risk of human exposure to mosquitoes. Recent studies have emphasized the importance of considering these factors when estimating the geographic-specific impact of vector control interventions and for implementing vector control strategies [4–8]. Additionally, the physical and chemical properties of the various interventions, such as the physical integrity and insecticide efficiency of ITNs and how each of these vary over time, also strongly impact the effectiveness of vector control [9–11].

Mathematical models of malaria transmission are frequently used to integrate quantitative evidence about the effects of malaria interventions to enhance prediction of impact and planning of interventions [12–14]. This type of modelling has become an important part of decision-making, in particular for guiding national malaria strategic plans in malaria-endemic countries [15–17]. For the models to accurately quantify the impacts of interventions, data from experimental hut trials and cluster-randomized control trials [9,18–26] are generally used to parameterize their effects [12,27–31]. Nonetheless, model parameterizations should also consider local variations in human behaviour and thus human exposure to mosquito bites. Considering human behavioural data and setting-specific differences in mosquito biting and bionomics can improve model predictions of intervention effectiveness [5].

Integrating human activity, mosquito biting patterns and other entomological characteristics to adjust the estimated impact of vector control interventions comes with its challenges. Many independent studies with different experimental techniques and data recording approaches are involved. Comprehensive data are rarely collected at the same location and time. Several existing models and studies account for the life parameters of mosquitoes estimated from entomological data and have combined information on mosquito biting and human activity [7,8,30,32–34]. However, these are only a few studies and have been limited to a handful of locations. A comprehensive framework collating the different data types, allowing for direct data integration and interfacing with models to estimate location-specific intervention impact in a systematic way has been lacking. Building on previous modelling of the mosquito feeding cycle [32] and of vector control impact [9,28,30], we have developed the AnophelesModel R package (Fig 1) to address these challenges.

## Design and implementation

AnophelesModel estimates the species- and geographic-specific impact of vector control interventions by integrating multiple layers of input data (Fig 1). Users can select input data on mosquito bionomic characteristics, mosquito and human activity patterns, and the effects of interventions (cf. Section A in S1 Text) from a curated database included in the package, or input their own data. The package employs these inputs to parameterize a mathematical model describing the mosquito feeding cycle [32] which infers how the state to state transitions within the feeding cycle are affected by different interventions, considering their decay over time (Box 1). Thus, the model estimates the reduction in vectorial capacity for a given

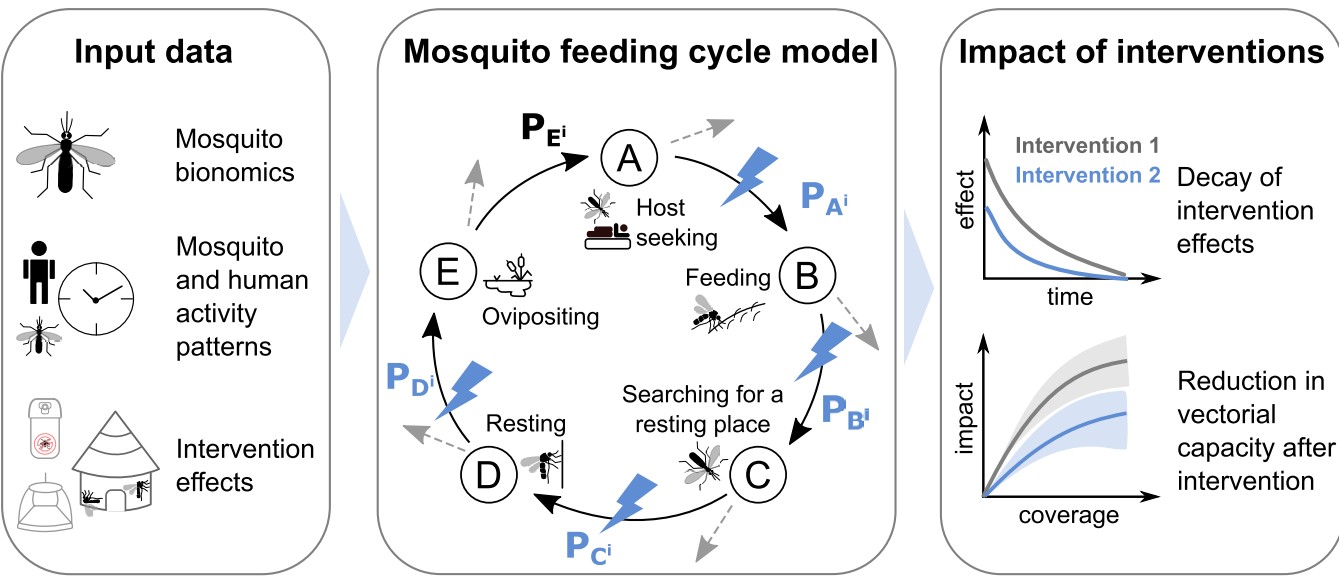

**Fig 1. Overview of the AnophelesModel R package and its components.** The package integrates various types of input data (first panel) to parameterise an existing model of the mosquito feeding cycle [32] (middle panel). This model represents the feeding cycle states with letters A through E and transition probabilities $P_A^i$–$P_E^i$ between consecutive states. The dotted, grey arrows indicate that mosquitoes can die at each stage. Blue lightning symbols indicate the probabilities affected by the vector control interventions included in the package. This model is used to infer the species-specific effects of vector control interventions, including the decay of these effects over time and their impact on vectorial capacity (third panel).

## Box 1. Key aspects of the mosquito feeding cycle

Every day, mosquitoes emerge from breeding sites. After mating, female mosquitoes enter a feeding cycle characterized by a set of key consecutive states (Fig 1 middle panel). First, mosquitoes begin their search for a host to take a blood meal from, entering the "Host Seeking" state A. During this search, they can either die or land on a host of type *i* (human or animal) and start feeding, transitioning to the "Feeding" state B with probability $P_{A^i}$. While feeding, they can either die, or successfully complete their blood meal and start searching for a resting place, moving to state C with probability $P_{B^i}$. They can die while searching for a resting place, or find one and enter the "Resting" state D with probability $P_{C^i}$. If they survive resting, they return to the breeding site to lay their eggs, entering the "Ovipositing" state E with probability $P_{D^i}$. Surviving oviposition, they re-enter the cycle and start searching for a host again with probability $P_{E^i}$. Each vector control intervention affects mosquitoes at specific stages in the feeding cycle (Fig 1 middle panel).

Several field-measurable bionomics parameters provide insights into the success of mosquitoes in completing their lifecycle:

- Parous rate: proportion of host-seeking mosquitoes that have laid eggs at least once.

- Human blood index: proportion of blood meals that mosquitoes obtain from humans.

- Sac rate: proportion of mosquitoes that laid eggs on the same day.

- The preference of mosquitoes to rest and feed indoors, known as endophily and endophagy, respectively.

intervention. The vectorial capacity is defined as the total number of subsequent infectious mosquito bites originating from each mosquito biting a human infected with malaria. The package allows the user to run analyses for interventions and species-bionomics with self-provided data. It can compare multiple interventions in terms of their effect on vectorial capacity for various mosquito species across a range of geographical settings. Furthermore, it produces ready-to-use outputs which can be plugged into established models of malaria transmission dynamics such as OpenMalaria [35,36].

## Mosquito bionomics data

The feeding cycle model relies on quantified ecological and bionomic characteristics of the mosquitoes (Box 1). AnophelesModel allows the user to input their own data and tailor the entomological model to the vector species of interest. Additionally, it also harbours an extensive database of relevant parameters collated from published literature and publicly available sources. Using a Bayesian hierarchical model applied to previously-published entomological data [30,37,38], mosquito bionomic parameters were derived for *57 Anopheles* species and 17 complexes (groupings of sibling species) and included in the package.

## Integrating mosquito and human activity patterns, estimating human exposure to mosquitoes

AnophelesModel implements a novel approach which allows use of input data on biting rhythms and human activity to adjust the effects of vector control interventions depending on the exposure of humans to mosquito biting, endophily (the proportion of indoor resting mosquitoes), and endophagy (the proportion of indoor feeding mosquitoes). Precisely, the deterrency, pre-prandial and post-prandial killing effects of the interventions are scaled by the corresponding setting-specific exposure coefficient. A detailed description of this approach is provided in Sections B and C in S1 Text. AnophelesModel also includes ready-to-use data on biting rhythms and human activity recently compiled by Sherrard-Smith *et al.* [7]. In addition, the package database contains entries from a non-systematic sample of publications [39–48].

## Modelling the effects of vector control interventions on the mosquito feeding cycle

The protective effects of vector control interventions used in the AnophelesModel package are defined in terms of the reduction in the proportion of mosquitoes reaching each stage in the feeding cycle (Fig 1 middle panel, Section B in S1 Text). There are three main effects modelled:

- Deterrency: the reduction in the availability rate of humans to mosquitoes per day, estimated based on the proportion of mosquitoes that fail to reach a protected human or are deterred from biting due to intervention

- Pre-prandial killing: the proportion of mosquitoes that are killed before feeding

- Post-prandial killing: the proportion of mosquitoes that are killed after feeding

The user can directly input these effects and use the package to conduct impact analysis for the interventions of their choice. In addition, a couple of parameterisations for intervention effects are already available in the package for long-lasting insecticide-treated nets (LLINs), indoor residual spraying (IRS) and house screening. These effects have been estimated using previously published intervention models (Table A in S1 Text). Accordingly, they have been parameterised with data generated from experimental hut trials and adjusted according to the

intervention-specific temporal decay functions, measuring attrition, change in use, insecticide decay and physical deterioration for LLINs, and insecticide decay for IRS [9,28,30].

Each intervention is assigned a duration corresponding to the time between consecutive deployments (e.g., 3 years for LLINs and 0.5 years for IRS). The effects and the resulting reduction in vectorial capacity are calculated for a finite number of equally spaced time points throughout this duration (denoted as interpolation points in the package). All intervention effects are adjusted for the exposure of humans to mosquitoes as described in the previous section. Currently the package does not include parameterizations for interventions that reduce the number of emerging mosquitoes (e.g., larviciding).

**Modelled effects of LLINs included in the package.** A previously published system of logistic regression models [9,30] can be used with the package to estimate the effects of LLINs deployments. The decay of physical properties of mosquito nets in terms of attrition, use, physical and chemical integrity has been estimated using the data from the President Malaria Initiative (PMI) net durability studies [49], and on data from Morgan *et al.* [50] as described in *Briet et al.* [9]. These datasets, containing properties of various net types in different countries, are also included in the package.

**Modelled effects of IRS included in the package.** The package includes several parameterisations of IRS effects for different insecticide and vector species combinations derived using experimental data from previous studies [23–26,51].

**Effects of house screening included in the package.** The effect of house screening interventions available in the package is assumed to be a linear relationship with the availability of humans to mosquitoes, with a 59% reduction as estimated in [30] based on data from Belize [52] and Ghana [53].

## Entomological model of the mosquito feeding cycle and vectorial capacity

The three previously defined types of inputs are used in the package to parameterize an entomological model of the mosquito feeding cycle published in [32]. Briefly, the model quantifies the probabilities of mosquito survival across five stages of the feeding cycle: host seeking, feeding, searching for a resting place, resting, and ovipositing (Box 1). The total numbers of host seeking, infected and infectious (sporozoite positive) mosquitoes are modelled through a system of difference equations with one-day time steps. In the absence of intervention pressure, the stage-specific survival probabilities are assigned the values derived in *Chitnis et al.* [32]. Intervention effects are modelled through reductions in these probabilities. The vectorial capacity is calculated analytically using the formulation derived in *Chitnis et al.* [32] and constitutes a proxy for the intervention impact, the main output of the AnophelesModel package.

## Interfacing with models of malaria transmission dynamics

The AnophelesModel package generates vectorial capacity estimates that can be seamlessly integrated into any compartmental model of malaria dynamics. This approach has been successfully utilized in previous studies, such as: [30,54,55]. In addition, the package estimates the decay of intervention effects over time and generates parameterizations of vector control components which may be used for running simulations with the OpenMalaria model. Specifically, AnophelesModel contains functions for producing formatted entomology and intervention input for the OpenMalaria model. OpenMalaria is an agent-based, stochastic model of malaria transmission dynamics, and it has been extensively described in previous publications [13,35,36,56]. It can be used to simulate malaria transmission within a population of individuals, deploy interventions and estimate their impact on malaria burden over time.

## Results

To illustrate the functionalities of the package, we provide examples using the data included in the package for two mosquito species, namely *Anopheles farauti* and *Anopheles gambiae* and compare the effects of the interventions available in the package. All the code used in the analysis presented in this paper is included in the package GitHub repository (see section Availability and Future Directions).

### Visualising human, mosquito and intervention characteristics

The AnophelesModel package can provide visualisations of the entomological characteristics of mosquito species at different locations and model how these impact various vector control interventions. One resource included in the package is a readily available database encompassing human activity patterns, mosquito biting patterns, mosquito entomological characteristics and intervention characteristics. The user can directly access the various data types through dedicated data objects. A detailed description of these data objects is provided in the package documentation.

The package database can be queried, for example to analyse how *An. gambiae*, among the dominant malaria vectors in sub-Saharan Africa [57], differs from *An. farauti*, a major vector in Papua New Guinea (PNG) (Fig 2). The two species are different not only in their bionomics, but also in terms of their biting patterns. *An. gambiae* has higher parous rates, sac rates, and human blood index, and is more endophagic than *An. farauti* (Fig 2A). Furthermore, *An. gambiae* preferentially bites indoors during the night, while *An. farauti* also bites outdoors, especially in the early evening (Fig 2B). These differences all affect the modelled impacts of interventions such as LLINs. In the following example, we demonstrate how AnophelesModel can be used to compare the impacts of LLINs, as well as other vector control measures, for these two species in their respective settings mainly relying on the data present in the package database, and incorporating new, recently published data on human behaviour for a PNG-like setting [58] (Fig 2C).

### Quantifying and comparing the species-specific impact of vector control interventions

First, we used AnophelesModel to incorporate the different mosquito, human and intervention data (Fig 2) accounting for the uncertainty in mosquito bionomics estimates (Section D in S1 Text) and to model the effects of LLINs for the two species using distinct values for deterrency, pre- and post-prandial killing effects for the two settings. We estimated higher effects of LLINs for *An. gambiae* in the Kenyan-like setting compared to *An. farauti* in the PNG-like setting (Fig 3A), and a correspondingly higher reduction in vectorial capacity for *An. gambiae* in the Kenyan setting (Fig 3B). In a second analysis, for the same mosquito species, we compared the estimated impact of LLINs to other interventions available in the package, namely IRS, house screening, and a combination of LLINs and IRS (Fig 4). For all interventions, the effects for *An. gambiae* were superior to those for *An. farauti*. Sections E and F in S1 Text show how this analysis can be extended to OpenMalaria and to combinations of several mosquito species.

The effectiveness of a vector control intervention is influenced by both its chemical and physical properties, and by the alignment of its temporal effects with the circadian rhythms of human behaviour and the mosquito biting patterns. With human presence indoors and in bed exhibiting the patterns shown in Fig 2C, a substantial proportion of the bites from *An. farauti* occur in the early evening when people are not yet sleeping under a net, in contrast to *An.*

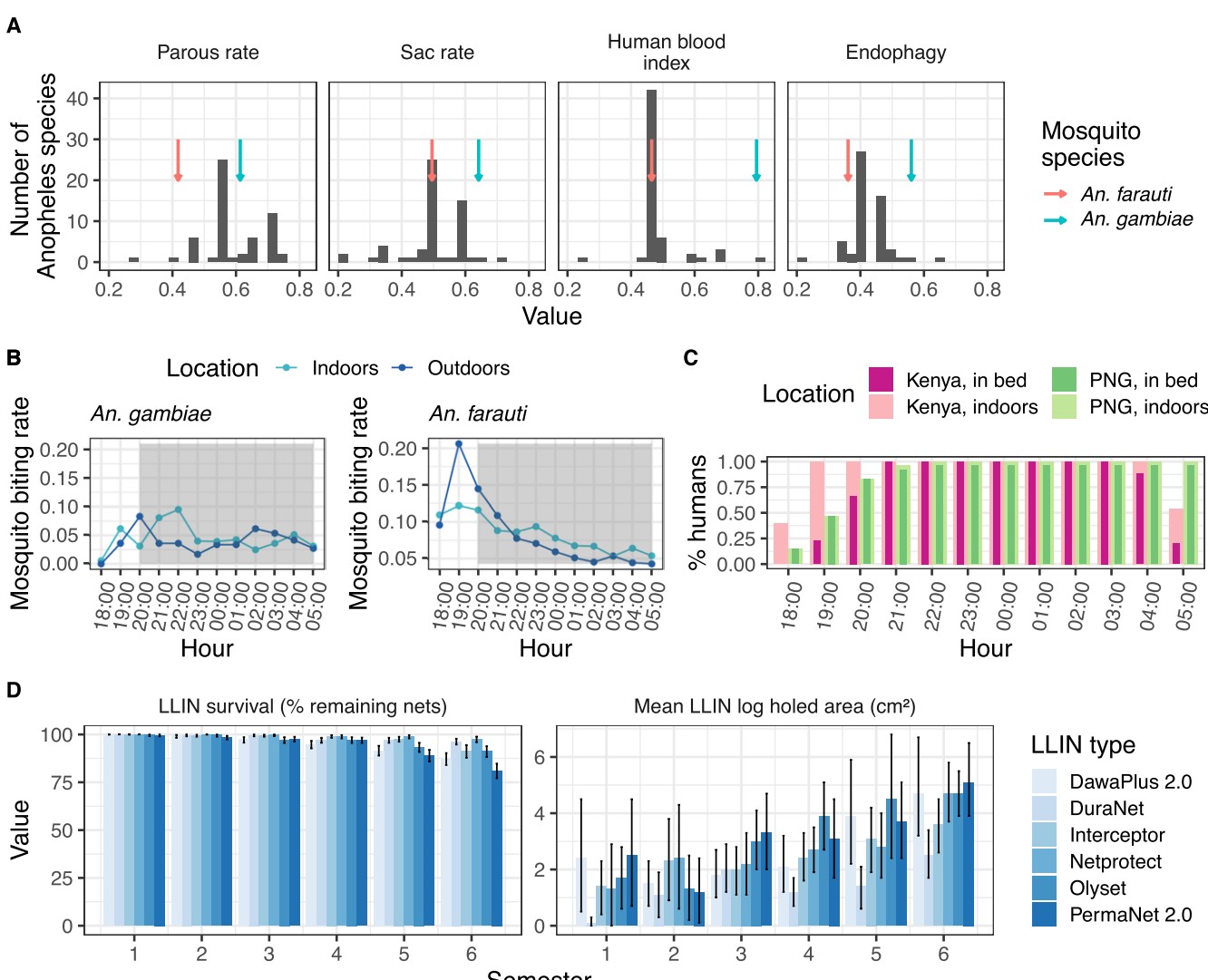

**Fig 2. Examples of the key types of data available within the AnophelesModel database which can be used to estimate the impact of vector control interventions.** The package provides **(A)** entomological parameters, **(B)** mosquito biting patterns, **(C)** human activity patterns, and **(D)** intervention properties, which can be used to parameterise an entomological model of the mosquito feeding cycle. Examples are provided for *An. gambiae* and *An. farauti* in Kenya and Papua New Guinea (PNG) settings, respectively. In panel **(A)**, the arrows indicate the bars corresponding to the two mosquito species. In panel **(B)**, hourly biting rates are shown for both species, while the grey area highlights the time when people sleep under a net. Panel **(C)** displays the hourly proportions of people located indoors and outdoors for each geographical setting. Panel **(D)** summarizes the observed variation in physical properties of LLINs in a Kenya-like setting [9] for each semester (every 6 months) during 3 years. Data sources of all data types are specified in the "Design and Implementation" section.

*gambiae*, which mostly bites at night. Thus, as found in previous analyses of the African data [7], the mosquito and human activity patterns strongly affect the estimated impact of vector control interventions, even when the physical and chemical durability of the mosquito nets are uniform (Fig 2D).

Similar to the examples provided for *An. gambiae* and *An. farauti*, the AnophelesModel package can be used to estimate and compare how the effects of interventions vary for other mosquito species and geographical locations. The user is not limited to the package database, but can input new data and use these in the modelling. The package documentation provides further examples illustrating the use of new data and also reproducing previously published analyses comparing *An. gambiae* and *An. albimanus* [30].

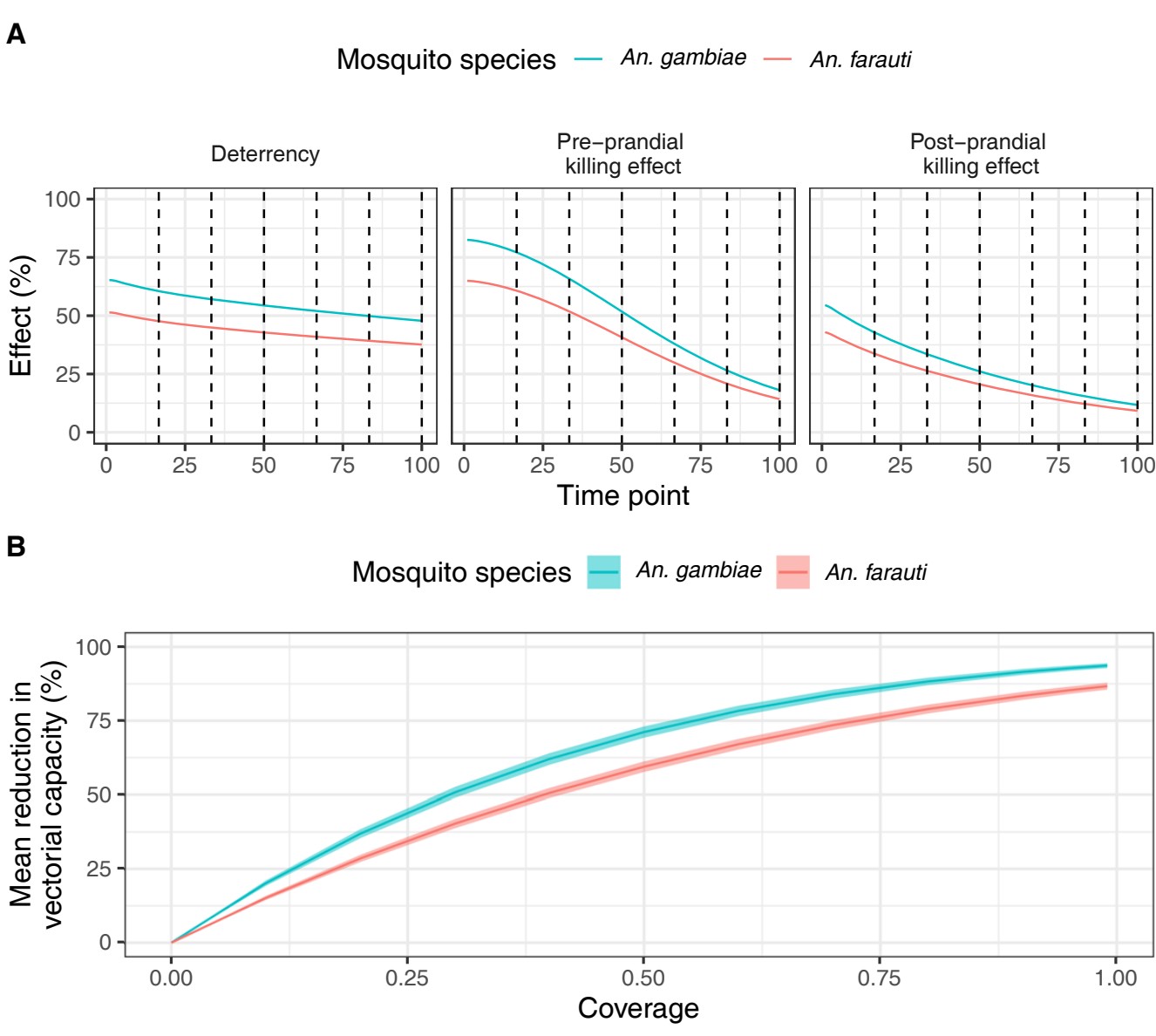

**Fig 3. Estimated effects of LLINs deployment for *An. gambiae* and *An. farauti*.** Mosquito, human and intervention data are combined in the AnophelesModel package to estimate the different types of intervention decay throughout time **(A)**, as well as the resulting mean reduction in vectorial capacity for varying LLINs deployment coverages (here equivalent to LLINs usage) **(B)**. The time units in panel **(A)** are defined by 100 equally distanced interpolation points across the duration of the interventions (i.e., 3 years for LLINs, with dotted lines marking each semester). The ribbons in panel **(B)** correspond to the variation of the vectorial capacity estimated based on the confidence intervals of the mosquito bionomics parameters (details on uncertainty propagation provided in Section D in S1 Text).

## Availability and future directions

The AnophelesModel R package source code and data are publicly available online in a dedicated GitHub repository at https://github.com/SwissTPH/AnophelesModel. A user-friendly website available at https://swisstph.github.io/AnophelesModel/index.html provides package installation instructions, comprehensive descriptions of functions, parameters and data, and detailed examples of use-cases. A systematic tutorial and documentation of the different package functions are provided at https://swisstph.github.io/AnophelesModel/articles/

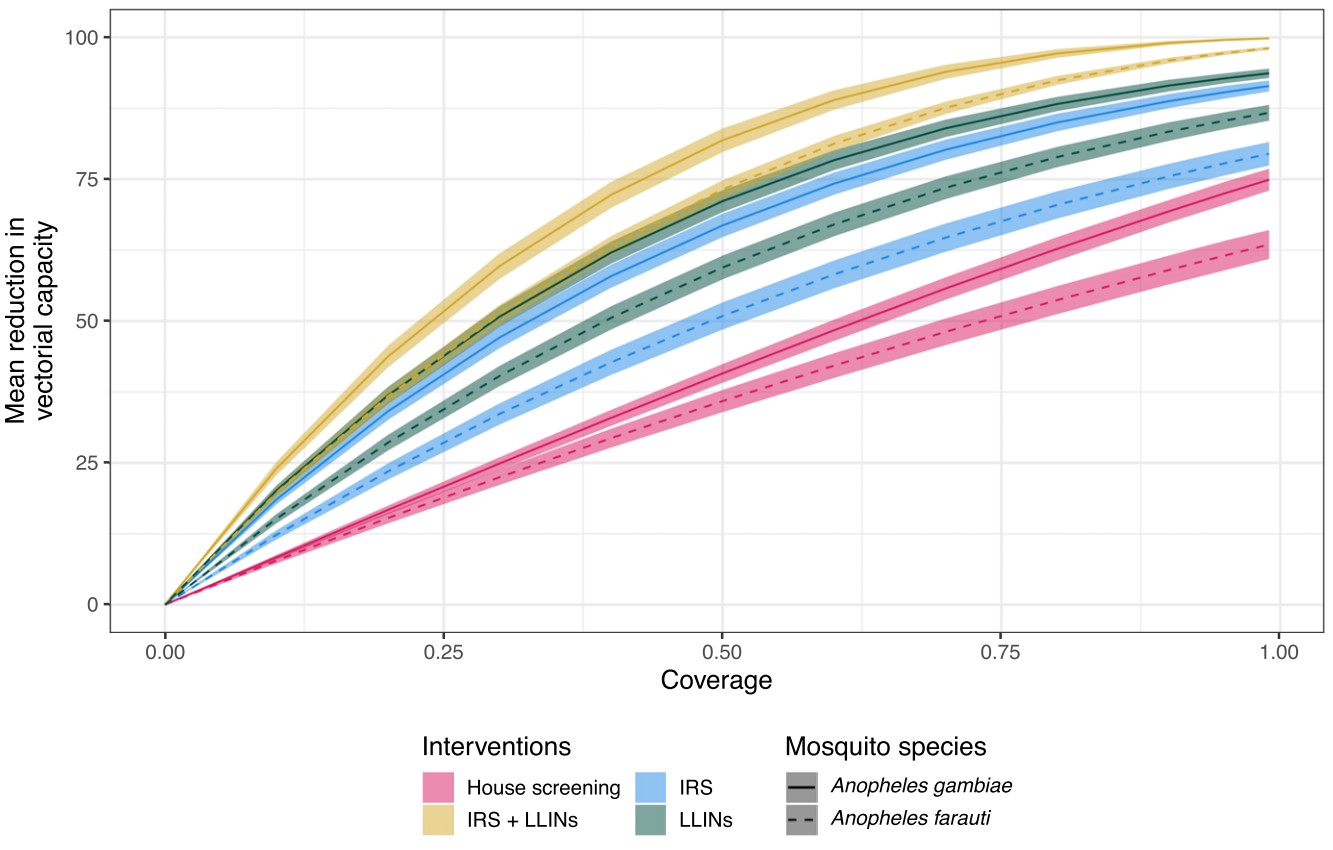

**Fig 4. Estimated impact of various interventions for *An. gambiae* (solid curves) and *An. farauti* (dashed curves).** The impact was estimated for the three types of interventions available in the package (IRS, LLINs and House screening), as well as for the combination of IRS and LLINs. The ribbons correspond to the variation of the reduction in vectorial capacity estimated based on the confidence intervals of the mosquito bionomics parameters.

AnophelesModel.html. Furthermore, all code used for the examples presented in this paper and for generating the corresponding figures is available at https://github.com/SwissTPH/AnophelesModel/tree/main/extdata.

Patterns of human exposure to mosquitoes alongside mosquito bionomics should always be considered when using impact modelling to make decisions about vector control options in different geographical settings [4,5,7]. For this purpose, the AnophelesModel package combines these different types of data to provide inputs into malaria models. In this paper, we have provided an example describing how to use the package outputs with the OpenMalaria model [35,36]. In the presented analysis, following inclusion of the exposure-adjusted intervention effects in OpenMalaria, we observed a clear difference in public health impact of LLINs deployment between the Kenya-like and PNG-like settings with the same level of pre-intervention transmission prevalence.

The value and usability of the package, as well as its interfacing with other models, have been already demonstrated in other published applications. For example, in a recently published study, AnophelesModel was used to inform the impact of vector control in a compartmental model of *Plasmodium vivax* malaria dynamics applied to identify malaria transmission hotspots in Panama [54]. Furthermore, the package has been incorporated in a mathematical

modelling framework to quantify the country-specific impact of interventions against *Plasmodium vivax* malaria [55].

The AnophelesModel package is flexible beyond the provided database, allowing the user to plug in new data and parameters and model intervention effects for a custom setting. The package database is not exhaustive and does not account for seasonal variation or variation by human age or occupational group. The package is a powerful tool for exploring how the impact of vector control interventions changes following the observed variation in input mosquito biting and human behaviour patterns. It can help parameterize various intervention models for locally specific conditions in trial settings, and thus check and validate the performance of these models in simulating the trial results. This can be done using malaria survey data or randomized control trial data, as demonstrated in previous studies for several intervention models included in the package [59,60]. The package informs the user about the availability or absence of previous validation whenever the user selects an intervention model from the database.

Planned developments of the AnophelesModel package include extension of the database of mosquito, human behaviour and intervention characteristics through systematic reviews, including more recently-generated data and intervention models. Currently three interventions are modelled within the package, namely IRS, LLINs and house screening, but other interventions such as spatial repellents and attractive toxic sugar baits will be added in the future.

## Supporting information

**S1 Text. Supplementary methods, figures, and tables.**
(DOCX)

## Author Contributions

**Conceptualization:** Monica Golumbeanu, Olivier Briët, Clara Champagne, Jeanne Lemant, Munir Winkel, Marianne Sinka, Nakul Chitnis, Tom Smith.

**Data curation:** Monica Golumbeanu, Olivier Briët, Barnabas Zogo, Marianne Sinka, Tom Smith.

**Formal analysis:** Monica Golumbeanu, Olivier Briët, Emilie Pothin, Tom Smith.

**Funding acquisition:** Melissa Penny, Emilie Pothin, Tom Smith.

**Investigation:** Monica Golumbeanu, Olivier Briët, Clara Champagne, Jeanne Lemant, Munir Winkel, Barnabas Zogo, Maximilian Gerhards, Nakul Chitnis, Emilie Pothin, Tom Smith.

**Methodology:** Monica Golumbeanu, Olivier Briët, Clara Champagne, Jeanne Lemant, Munir Winkel, Maximilian Gerhards, Emilie Pothin, Tom Smith.

**Project administration:** Monica Golumbeanu, Tom Smith.

**Resources:** Monica Golumbeanu, Nakul Chitnis, Melissa Penny, Emilie Pothin, Tom Smith.

**Software:** Monica Golumbeanu, Olivier Briët, Clara Champagne, Maximilian Gerhards, Emilie Pothin, Tom Smith.

**Supervision:** Nakul Chitnis, Melissa Penny, Emilie Pothin, Tom Smith.

**Validation:** Monica Golumbeanu, Jeanne Lemant, Maximilian Gerhards, Emilie Pothin, Tom Smith.

**Visualization:** Monica Golumbeanu, Emilie Pothin, Tom Smith.

**Writing – original draft:** Monica Golumbeanu, Tom Smith.

**Writing – review & editing:** Monica Golumbeanu, Olivier Briët, Clara Champagne, Jeanne Lemant, Munir Winkel, Barnabas Zogo, Maximilian Gerhards, Marianne Sinka, Nakul Chitnis, Melissa Penny, Emilie Pothin, Tom Smith.

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
