## [Decision Letter · Decision Letter 0]

10 Apr 2024

Dear Dr Golumbeanu,

Thank you very much for submitting your manuscript "AnophelesModel: An R package to interface mosquito bionomics, human exposure and intervention effects with models of malaria intervention impact" for consideration at PLOS Computational Biology.

As with all papers reviewed by the journal, your manuscript was reviewed by members of the editorial board and by several independent reviewers. In light of the reviews (below this email), we would like to invite the resubmission of a significantly-revised version that takes into account the reviewers' comments.

I'll briefly note that all three reviewers were positive and recommended minor revisions. However, I've chosen to mark this as "major revisions" simply because the three sets of recommendations are generally complementary to each other. There is no meaningful difference, but I feel that this better reflects the likely effort in revising the paper. 

We cannot make any decision about publication until we have seen the revised manuscript and your response to the reviewers' comments. Your revised manuscript is also likely to be sent to reviewers for further evaluation.

Sincerely,

Daniel B Larremore, Ph.D.

Academic Editor

PLOS Computational Biology

Natalia Komarova

Section Editor

PLOS Computational Biology

Editor's synthesis: all three reviewers were positive and recommended minor revisions. However, I've chosen to mark this as "major revisions" simply because the three sets of recommendations are generally complementary to each other.

Reviewer's Responses to Questions

**Comments to the Authors:**

Reviewer #1: Please see the uploaded comments

Reviewer #2: Review is uploaded as a Microsoft Word document.

Reviewer #3: The manuscript introduces an R package specifically crafted to process input data on vector bionomics, facilitating the generation of species-specific impacts on the carrying capacity of vector control interventions. It stands as a solid, well-written manuscript, and fits well within the broader scientific context of malaria modeling and vector research.

The R package offers a valuable contribution to the field, particularly in its interaction with a precompiled, or user-defined, database allowing to project impact on vectorial capacity for different mosquito species and vector control interventions.

In consideration of a wider malaria research community, and to enhance its value beyond OpenMalaria users, the manuscript could highlight the functionality of its three main components more prominently.

The core functionality of the model is the estimation of species and geographic-specific impact of vector control interventions, which is described in main text and supplementary file. This direct output of the software is illustrated in Figure 2. Conversely, the simulated health impact, which stems from external malaria model simulations (depicted in Figure 3), is not. The authors might therefore consider moving Figure 3 and description of OpenMalaria to the supplement, to allocate more space for elaboration on the various options for estimating intervention impact. For instance, including all three interventions available, such as IRS and house-screening.

Relatedly, in the current format, the manuscript might also inadvertently imply that the ready-to-use outputs could be utilized for established transmission models, although this applies solely to OpenMalaria. Hence the title and heading might appear overstated.

Suggestions for improvements the authors might consider include:

1) Elaborate more on the generic utility of the model, (i,e, how in R to interact with the functionalities described in the supplement , how to work with the input data, or description of effects generated for the two other types of interventions available) would be of greater interest for the broader community.

2) Move the example "Interfacing AnophelesModel with models of intervention impact" to the supplement, as the simulated health impact is not a direct output of the software presented in the Article. The space can instead be used to address 1)

Some minor suggestions and comments:

- Aligning or integrating the flowchart presented in the manuscript with the programmatic flowchart outlined in the R package documentation. Notably, in the manuscript intervention effect is included as ‘data’ (which one might interpret to be an input), whereas in the package documentation it is presented as a part of the entomological model component.

- The methods section would benefit from adhering to the same structural framework as depicted in the flowchart (Figure 1),i.e. describing the three data types first, then the Entomological model of the mosquito feeding cycle and vectorial capacity

- Figure 2D and Fig 3: A) both describe the LLIN decay however they use different units and description in the x-axis, and the text provides an example of decay in years. This could be considered to be aligned, i.e. using semester also in Fig 3A.

- Figure 4: The Kenya and PNG settings in the figure are missing the ‘like’. Given the high heterogeneity of vector dynamics within a country using the region names would be (i.e. Kisumu) would be clearer. For most clarity, author might consider labelling the outcome directly by the vector species, since only one vector species was used while in geographic areas often multiple vectors occur. In the two simulations for Kenya and PNG, the input EIR was set to the same level, not ‘similar’ (i.e. L286, L311)

- The inclusion of syntax and output examples would be beneficial for this type of article.

- It's important to address how insecticide resistance is considered, or can be factored in when utilizing the package to obtain intervention effect sizes. While experimental hut trial and cluster randomized trial data are used to parameterize effects, it's crucial to clarify whether resistance levels in the area are accounted for in the analysis, or whether all effects generated apply to fully susceptible mosquitoes.

- A potential user of the package might also be interested in whether the vector model can produce intervention effect for the combinations of LLINs and IRS , which likely affets the probabilities in host-related entomological parameters hence the reduction in vectorial capacity?

**Have the authors made all data and (if applicable) computational code underlying the findings in their manuscript fully available?**

Reviewer #1: Yes

Reviewer #2: Yes

Reviewer #3: None

PLOS authors have the option to publish the peer review history of their article (what does this mean?). If published, this will include your full peer review and any attached files.

Reviewer #1: **Yes: **Ellie Sherrard-Smith

Reviewer #2: No

Reviewer #3: No
---

## [Decision Letter · Decision Letter 1]

16 Aug 2024

Dear Dr Golumbeanu,

We are pleased to inform you that your manuscript 'AnophelesModel: An R package to interface mosquito bionomics, human exposure and intervention effects with models of malaria intervention impact' has been provisionally accepted for publication in PLOS Computational Biology.

Best regards,

Tobias Bollenbach

Section Editor

PLOS Computational Biology

Reviewer's Responses to Questions

**Comments to the Authors:**

Reviewer #2: All recommended revisions have been adequately addressed.

Reviewer #3: The reviewer thanks the authors for their careful and thorough revisions and responses to the reviewers' queries. All comments have been sufficiently addressed, and I have no further remarks on this much-improved version of the manuscript.

**Have the authors made all data and (if applicable) computational code underlying the findings in their manuscript fully available?**

Reviewer #2: Yes

Reviewer #3: Yes

PLOS authors have the option to publish the peer review history of their article (what does this mean?). If published, this will include your full peer review and any attached files.

Reviewer #2: No

Reviewer #3: No

---

## [Editor Report · Acceptance letter]

7 Sep 2024

PCOMPBIOL-D-23-01675R1 

AnophelesModel: An R package to interface mosquito bionomics, human exposure and intervention effects with models of malaria intervention impact

Dear Dr Golumbeanu,

I am pleased to inform you that your manuscript has been formally accepted for publication in PLOS Computational Biology. Your manuscript is now with our production department and you will be notified of the publication date in due course.

With kind regards,

Zsofia Freund
